# Drivers of Solid Waste Segregation and Recycling in Kampala Slums, Uganda: A Qualitative Exploration Using the Behavior Centered Design Model

**DOI:** 10.3390/ijerph191710947

**Published:** 2022-09-02

**Authors:** Richard K. Mugambe, Rebecca Nuwematsiko, Tonny Ssekamatte, Allan G. Nkurunziza, Brenda Wagaba, John Bosco Isunju, Solomon T. Wafula, Herbert Nabaasa, Constantine B. Katongole, Lynn M. Atuyambe, Esther Buregyeya

**Affiliations:** 1Department of Disease Control and Environmental Health, School of Public Health, College of Health Sciences, Makerere University, Kampala P.O. Box 7072, Uganda; 2Department of Public Health, Kampala Capital City Authority, Kampala P.O. Box 7072, Uganda; 3Environmental Health Department, Ministry of Health, Plot 6, Lourdel Road, Nakasero, Kampala P.O. Box 7272, Uganda; 4Department of Agricultural Production, College of Agricultural and Environmental Sciences, Makerere University, Kampala P.O. Box 7062, Uganda; 5Department of Community Health and Behavioral Sciences, School of Public Health, College of Health Sciences, Makerere University, Kampala P.O. Box 7072, Uganda

**Keywords:** waste segregation, waste recycling, slums, solid waste management, Uganda

## Abstract

Solid-waste management is a challenge in many cities, especially in low-income countries, including Uganda. Simple and inexpensive strategies such as solid-waste segregation and recycling have the potential to reduce risks associated with indiscriminate waste management. Unfortunately, these strategies have not been studied and adopted in slums in low-income countries. This cross-sectional qualitative study, therefore, used the behavioral-centered design model to understand the drivers of recycling in Kampala slums. Data were coded using ATLAS ti version 7.0, and content analysis was used for interpreting the findings. Our findings revealed that the study practices were not yet habitual and were driven by the presence of physical space for segregation containers, and functional social networks in the communities. Additionally, financial rewards and awareness related to the recycling benefits, and available community support were found to be critical drivers. The availability of infrastructure and objects for segregation and recycling and the influence of politics and policies were identified. There is, therefore, need for both the public and private sector to engage in developing and implementing the relevant laws and policies on solid waste recycling, increase community awareness of the critical behavior, and create sustainable markets for waste segregated and recycled products.

## 1. Introduction

The rapid population growth in urban centers has continued to amplify the challenge of solid waste management at both global and regional levels. At the global level, the already high rate of solid waste generated is expected to increase by 70%, from 2.01 billion tons in 2016 to about 3.40 billion tons by 2050 [1]. Although Sub-Saharan Africa generated 174 million tons of solid waste in 2016, the rate of generation in the region is expected to triple by 2050, largely due to the rapid population sprawl that characterizes cities in many low-income countries [1].

Indiscriminate/poor solid waste disposal is associated with not only health but also environmental health risks [1,2]. Such environmental health risks include blockage of drainage channels, which often leads to flooding and the creation of breeding sites for vectors such as mosquitoes and flies. These risks escalate the transmission of infectious diseases such as cholera and diarrhea, whose episodes have for long been common in cities in low-income countries [3]. Moreover, indiscriminate solid waste management is also associated with loss of aesthetics, and the release of green-house gases such as methane and nitrous oxide which have a negative impact on climate change [2,4]. Addressing these challenges requires innovation and waste reduction, reuse, and recycling (3Rs) have for instance been recommended [5,6,7]. Waste reduction is the practice of preventing waste by decreasing or eliminating the number of materials initially used. Reuse refers to the practice of using a material over and over again in its current form. Recycling, defined as any recovery operation by which waste materials are reprocessed into products, materials or substances whether for the original or other purposes, has enormous economic and ecological benefits, and is efficient and sustainable compared to other solid waste management methods such as incineration and landfilling [8].

Reuse and recycling of waste materials can only be successfully achieved with appropriate waste segregation at the source. Waste segregation involves separating the different waste streams at the source in order to reduce the contamination of those with potential recycling value and reduce the quantities that go to sanitary landfills [9]. Moreover, segregation reduces the volume of contaminated waste [10,11], which reduces the time and cost for waste recovery, and the associated occupational hazards [12,13]

Solid waste segregation and recycling (SWSR) are known to increase the recycling rate by 84% and substantially improve solid waste management; however, SWSR is not a culture in many cities in low-income countries such as Uganda. In Kampala, which is the capital city of Uganda, over 28,000 tons of solid waste are generated every month. However, only 40% of the generated waste is safely managed, with the rest being indiscriminately disposed of [14]. The remaining uncollected waste is indiscriminately dumped in unauthorized places resulting into nuisances, environmental contamination and transmission of diseases. Recycling and reuse are uncommon, and the drivers of SWSR in Kampala have not been documented. This study therefore used a behavior centered design (BCD) theory [15] to explore the drivers of SWSR in the slums in Kampala. This theory can be used to understand the complexity of human behavior, and how people can change their behaviors to start waste segregation and recycling.

## 2. Materials and Methods

### 2.1. Study Area

The study was conducted in the informal settlements (slums) of Makerere 1 in Kawempe division, Bukoto in Nakawa division, Kasubi in Rubaga division, Wandegeya in Kawempe division and Kamwokya 1 in the Central division of Kampala, Uganda. Kampala is the Capital City of Uganda and the most populated urban center, with a total population of 1,507,080 [16]. The population growth rate for Kampala is high, and this resulted in drastic population increase from 774,241 in 1991 to 1,189,142 in 2002 [16]. The rapid population growth has resulted in growth of slums, increased waste generation, and challenges in waste management. Currently, Kampala has 57 slums which make up at least a quarter of the total city area and houses approximately 60% of the total city population [17]. These slums are characterized by poor housing conditions mainly because of informality and poverty, poor access to water, sanitation and hygiene (WASH), and poor solid waste management, among other challenges [17,18,19].

### 2.2. Study Design

The study employed qualitative methods through the use of focus group discussions (FGDs) and key informant interviews (KIIs). The study was guided by the “Behavior Centered Design (BCD) theory and model [15], in Figure 1, which highlights the BCD process on the outside and the theory of change on the inside.

The theory suggests that the right behavioral response is dictated by the behavior settings (physical, social and temporal context) in which individuals find themselves [20]. Within the behavior setting, an interaction between the environment (modified by an intervention) and psychological change in the target population (body and brain) results in the performance of the target behaviors, which results in changes to the state of the world. The BCD theory was considered ideal for this study since it draws from a range of disciplines such as social ecology, psychology, and social marketing practice [21], and has been recommended for studying WASH behaviors [22], food hygiene behaviors [23,24], solid waste management behaviors [15], and other behaviors [25]. According to Aunger and Curtis [15], the drivers of behavioral change include brain-related factors (including knowledge, risk, motives, reactions, and psychological trade-offs), body-related factors (characteristic traits and sensations), factors related to the settings where the behavior takes place (infrastructure, props, roles, routine and norms), and factors in the broader environment (the biological, physical and social environment and the wider context). The BCD theory is broad enough to classify all of the determinants described in other frameworks. Table 1 provides definitions of each BCD determinant adapted for SWSR.

### 2.3. Study Participants

Twenty-nine (29) KIs, including Environmental Health officers working with the Kampala Capital City Authority (KCCA), community leaders, market leaders, village health teams, community SWSR champions, and officials from Ministry of Water and Environment (MWE), the National Environment Management Authority (NEMA), waste recycling companies and waste collection companies were interviewed. The KIs were selected based on their knowledge and experience with solid waste management projects and activities in the City, and snowball sampling was used. Additionally, a total of 8 FGDs were conducted with market vendors working in the markets in study parishes and residents (male and females) in the residential settings in the study settings.

### 2.4. Data Collection

KIs were invited to participate in the study by the principal investigator through a phone call or email. Participants for the FGDs were invited to participate in the study by either a community health worker, community or market leader through a phone call or word of mouth a day before the interview. The FGDs were homogeneous, consisting of 8–10 people lasting at least 1 h. Of the eight FGDs conducted, four were with commercial vendors and the other four were with residents in the study area. During the discussion, an audio recorder was used to store the information being given and also notes taken by a note taker. Prior to the group discussions and KIIs, the participants were introduced to the purpose of the FGD and KI interview, respectively, and informed written consent was obtained.

Data collection was accomplished following FGD and KI interview guides, which were designed based on critical review of existing literature [9,13,15,26,27]. The data collection tools explored the participants’ views on management of solid wastes in the community, positive and negative aspects observed in relation to solid waste management in the community, waste management challenges (behavioral, social, economic, infrastructural challenges, effects and who is affected), drivers to SWSR, and touch points for delivering solid waste management messages in the community. Prior to data collection, pretesting of data collection tools was performed in similar residential and commercial settings in a different district, and essential adjustments in the tools were made.

### 2.5. Data Management and Analysis

All interviews were audio recorded, and the recordings in both the local language (Luganda) and English were transcribed verbatim. Translation of the transcripts from the local language to English was performed by experienced researchers, and two people read the transcripts and assigned meaningful units to each response which were later combined to form a code book. All data were entered into ATLAS ti software version 7.0 for coding. Analysis was carried out using directed content analysis, and the BCD model and checklist [15,26] were used to identify recurrent themes and subthemes by categorizing the codes.

## 3. Results

### 3.1. Background Characteristics of FGD Participants

About 44% of the respondents were aged between 30–39 years, more than half were female, and 50% had completed secondary education (Table 2).

### 3.2. Drivers of Waste Segregation and Recycling

The findings are organized and presented based on the five categories of drivers in the BCD checklist (Table 1), and the themes and subthemes are presented in Table 3. The categories of drivers include environment, brains, body, behavior settings and context.

#### 3.2.1. Environment-Related Drivers

##### Physical Environment

The siting and construction of houses in the study areas were poorly planned and hence not conducive for waste segregation. Houses were built very close to each other and thus had minimal space to store the different garbage storage containers or sacks. Moreover, some of the houses were built near drainage channels and in the road reserves, as stated by one of the key informants below;


*“The nature and size of places/homes where people live, they don’t have enough space to do the segregation of their waste since everyone is just next to the neighbor’s doors because of the nature of our housing.”*
P31: Key informant Kasubi

Refuse skips and collecting centers for segregated materials were available in the study areas, but there were very few, and they were not centrally placed to enable reachability for most users. The refuse skips were, therefore, mainly utilized by those who lived near them. Poor siting of the refuse skips did not only lead to the introduction of middlemen who carried the waste from homes that were located far away from the refuse skips but also created illegal waste dumping sites (these offered great convenience to the residents). This was mentioned by one of the key informants as below;


*“A person would not have thrown solid waste in the channel when the skip is near. A person would not throw solid waste at another person’s veranda when the skip is near but unfortunately the available skips are usually far away from the households. One considers the journey they have to take and sometimes does not have the 500 shillings (0.137 USD) to pay a child for delivering the waste to the refuse skip, and decides to dump it there. So, it affects us if refuse skips or segregation waste collecting centers are far, because of the high accumulation rate of solid waste in communities that are spread apart.”*
P28: Key informant Wandegeya

##### Social Environment

Some community members who were engaged in SWSR encouraged their neighbors and peers to do the same. For example, those making briquettes, which save on fuel/energy costs, were slowly influencing others in the community to adopt the use and making of briquettes, as one of the participants stated below;


*“We have tried to collaborate as neighbors to do certain things together like encouraging each person to have a collection sack next to their doors, we also separate charcoal dust and a few other wastes and reuse them for making briquettes. We also try to collectively pay for the waste as neighbors in order to reduce on the cost irrespective of the tempers and our differences.”*
P6: Participant FGD Kasubi-Residential

Community leaders also mentioned using a social network referral system whereby if a community member expressed interest in recycling waste, they were referred to other members already doing so for mentorship and apprenticeship. The referral system was used so as to encourage waste segregation, reuse and recycling, as mentioned below;


*“As the local authorities, we keep referring other families to those who make briquettes and have earned a living from them as a way of encouraging them to embrace segregation, reuse and recycling as a way of generating income.”*
P22: Local leader-Makerere 1

#### 3.2.2. Brains Related Drivers

##### Executive Brain

The target community was largely not aware of the benefits of SWSR. The respondents reported having poor knowledge and lack of training on the benefits of segregating waste. The few who segregated knew that they needed the waste to be used as animal feed or for making crafts such as mats and for monetary returns. This was reported in the FGDs as indicated in the quotes below:


*“...we hear that many things can be made of solid waste but in hearing that, we need to get trainings like on Bukedde TV. I saw that the maize remains are used in making flowers. They cooked together with the dye for making sisal and make different designs. We would be able to get things from wastes when we are taught, but how will I know without getting trainings on reusing or recycling solid wastes like bottles.”*
P34: FGD 3 Makerere 1 Residents

The community mostly knew and practiced segregation of food peelings for use as animal feeds while the remaining waste was disposed of unsegregated. Banana peelings were the most commonly sought-after waste in the community by those who mostly carry out zero grazing of cattle (a system where cattle are usually kept in a farm house, and farmers bring feed and water to the animals) in their homes. This was attested to by some respondents as below;


*“We don’t segregate this waste; we just mix everything together except for the peelings from food since it is collected by farmers for animal feeds.”*
P6: Participant FGD Kasubi

##### Motivated Brain

Segregation was seen as rewarding for particular waste types. Segregation of food peelings, for example, was mentioned as rewarding since, after proper collection (without mixing with other waste), it was sold off for monetary returns. More so, the availability of a ready market for the segregated food peelings also motivated the community to segregate them, as stated below;


*“Because they are sure if they collect banana and cassava peelings in one bag, people with animals will come looking for feeds, so they are able to sell to them and make money.”*
P8: Local leader-Kamwokya


*“What I see mostly motivating people are those that buy the plastic bottles from the people and recycle them. People fight for the plastic bottles and this encourages them to sort them out of other waste streams. The people having animals also encourage residents to sort the food peelings from the other garbage since they pay about Uganda shillings 500 (approximately 0.137 USD) per 200 liter container to get them from residents.”*
P35: Participant FGD Kamwokya


*“It comes to money. Let us say there is a weighing scale that buys plastics. People would not play around with plastics because someone has got what to do. Like how we hear that electricity comes from wastes, there comes, and area where they collect wastes from and buy it. I will relate it on money. For example, there is market for scrap metals and you cannot find them everywhere. So, when they put in money that will end.”*
P34: FGD 3 Makerere 1 Resident

In addition to the monetary returns expected from segregating waste, the provision of waste segregation equipment by potential buyers also prompted the community to segregate their waste as indicated below;


*“In addition, farmers dump their sacks in the families to get the food remains. Some of them have animals like pigs to feed while the others have plantations that need organic manure or biodegradable waste materials. This helps many people sort the rubbish and make sure they fill the sacks. Each sack (with a volume of about 200 liters) is Uganda shillings 500 (approximately 0.137 USD) so they fill it up in order to get the money.”*
P16: Key informant, Bukoto

The desire to live in a clean environment where children could play and keep safe without risk of becoming exposed to infectious diseases from the waste was mentioned as a motivating factor to segregate and recycle waste. Participants mentioned the need for supporting SWSR programs in their community so as protect their children against infections as indicated by one of the participants below;


*“...they should help us and bring those trainings on solid waste segregation. We should follow them and keep humanity by keeping the wastes so that it doesn’t make our children and us sick. We keep clean and protect ourselves, when we separate food remains bottles, broken plates and other items.”*
P3: Participant FGD Katanga

On the other hand, some participants were demotivated because of the low returns from the segregation process for some waste types compared to the effort put in. It was anticipated that those who segregated waste to make briquettes and mats received higher returns. This was illustrated as below;


*“There is a Chinese (in company Y) who said that all the plastic bottles and bags should be taken to him in Nakawa. A kilogram is at 100 shillings (0.027 USD). The road cleaners and sweepers worked hard to gather all these plastics and take for the Chinese but a woman would carry three bags of plastics and gets only 5000 shillings (1.368 USD). They got demotivated and stopped. Those that pick the plastics I am not sure how much they get from them. But the people making briquettes and straw mats get some good money from the garbage.”*
P24: Key informant Bukoto 1

Another demotivating factor for waste segregation was the indiscriminate loading of the waste on collection trucks at the point of collection. Respondents mentioned that even when they sort the waste, the collection truck operators load the waste on the trucks haphazardly, rendering their efforts to segregate waste useless, as attested to by a respondent below;


*“Why should I segregate yet when (company X) comes, it is going to come back and mix and also when the collectors don’t make the arrangements of where to take the segregated waste? There are some challenges there because you can only comfortably segregate when you know I am picking plastics and taking them to a known plastic recycling company. There will be that motivation.”*
P35 Key informant, private waste company

#### 3.2.3. Body Related Drivers

##### Socio-Demographic Characteristics

SWSR is mainly performed by institutions and the middle-income to rich people who have some education and reside in organized environments such as Kololo and Nakasero (Upper income residences). Respondents stated that these residences had better roads so waste collection trucks can easily move during waste collection. Moreover, these areas where reported to be having good solid waste storage containers and willingness to pay for garbage collection services (UGX 30,000 per month = 8.21 USD per month) which is high for the low-income people, as indicated below;


*“Yeah, this (waste segregation) is mostly done in organized institutions like schools, hospitals, housing estates, barracks and other settlement patterns like Nakasero and Kololo. People do because they are educated, have access to infrastructure within their settings, and they can afford to pay for the segregation bins and others waste management services.”*
P10: Key informant waste collection company

##### Human Senses

Participants mentioned that waste could easily be segregated if it is free from bad odors, especially under circumstances where there is no mixing of the different waste streams. Participants also mentioned that the mixture of food waste with other items such as disposable diapers results in an offensive smell. One of the underlying reasons for the non-segregation of the waste was mentioned as a lack of waste collection equipment for point of generation segregation, which consequently results in mixing different waste streams and smells. This was re-echoed by key informants who stated;


*“The problem we find from wastes is the smell. For example, a person may slaughter their hen, collects the blood, feathers and intestines in a polythene bag, and dumps the bag with other waste streams in a residential setting. Segregation of solid waste at source is a problem and people mix pampers, pads, plastic bottles, food remains and other waste streams. Removing useful products from such mixtures is difficult because of the smells that arise from the storage containers or sacks.”*
P34: Participant FGD 3 Makerere 1 Residents

##### Capabilities

Community members believed that SWSR requires special skills and is only practiced by those who have had the training as mentioned below;


*“We have not really considered segregating waste except for some few people who have attended some training workshops, and they have been making animal feeds from some waste streams but the rest is discarded in sacks as a mixture irrespective of the differences in the kinds of waste...”*
P6: Participant FGD Kasubi

More so, the community expressed the need to be trained, especially in waste recycling, so that they can team up and utilize the waste available to make different products and money, as indicated below;


*“...if we have people that can teach us waste recycling innovations, e.g., if they tell us you can recycle plastics in a certain explained way, we can group up as women in the community and create our own businesses to improve the management and recycling of the waste that surrounds us.”*
P33: Key Informant Kasubi zone 2

Participants revealed that they did not know much about what and how to segregate the solid waste and what to do with the sorted waste. Even with those that knew how to segregate and recycle waste, body weaknesses and laziness were cited as key impediments, as indicated below;


*“Some people are ignorant about it and others that know about recycling are weak or lazy to do it. They would desire to recycle the rubbish even under instructions but their bodies are weak. The briquettes you see require time to make them from the banana peelings.”*
P12: Key informant Bukoto

#### 3.2.4. Behavior Setting Drivers

##### Stage

Security of the waste segregation and disposal bins within the community was mentioned as a great concern, and areas with low crime rates and fewer cases of theft were reported to be more conducive for waste segregation and recycling since the storage containers would be safe. One of the key informants mentioned that:


*“Areas with low crime rates and fewer cases of theft are most likely to have better waste segregation and recycling rates. In areas with high crime rates, urban dwellers sometimes steal the storage/segregation sacks and pour rubbish in front of the owner’s home. This is because the sacks have monetary value. The sack alone costs 500 Shillings (approximately 0.137 USD) which is quite lucrative for the people in the slum community.”*
P19: Key Informant, government parastatal

Another behavior mentioned by the respondents was about some community members who did not dispose of waste in the appropriate bins and in a correct manner. The majority of the community members rather did not segregate at the source but dumped the waste anywhere near the waste bins leaving it for the collectors to sort and dump it as mentioned below;


*“...We have bins that we put on the road side for the pedestrians having yoghurt or soda to dump the rubbish there. But a person leaves home at 5 a.m. before KCCA starts working and they dump their garbage in the small bin. Sometimes they dump the large garbage next to the bin because they are sure that KCCA will come and collect the garbage from the bin. Those are some of the challenges KCCA faces. But the markets have these bins.”*
P24: Key informant Bukoto

##### Roles

The roles of individuals in waste segregation were not well known to the respondents, especially in residential settings. In the commercial settings, the roles of the shop and/or market attendants were quite better understood. The commercial areas were better managed because of the strong leadership that emphasized responsibility amongst members and the ‘punishments/fines’ attached to indiscriminate disposal of the rubbish. Unfortunately, the lack of proper guidelines in commercial settings affected segregation, and organic waste was taken away on a regular basis. Paper waste was recycled because it was considered clean, and some plastic collection was also performed by individuals and companies that needed it. In many settings, the only clear role for the community was to contain waste in their premises awaiting collection and take the waste to the collection cars key informant interviews though some participants mentioned using middlemen to transport the waste which in most times did not reach the collection car. More so, some community members detested paying waste collection fees indicating that it should be a free service and hence many kept dumping indiscriminately as illustrated below;


*“There are people who pay solid waste collection fees, ranging between 3000–5000 Uganda shillings (0.821–1.367 USD) and waste collection companies spend more than 1 month without showing up at all. Companies that buy segregated waste items can also take long without showing up in communities where some waste segregation happens. What follows are the kids who keep lying that the garbage collectors have come and once you give them some money like 1000 Uganda shillings (0.274 USD) for waste disposal, they just turn the next corner and abandon the solid waste at the neighbor’s place and disappear. Others detest paying for garbage on top of the rent they remit thus keeping dumping the rubbish in other places in smaller portions of polythene”*
P32: FGD participant Kasubi

The community viewed the job of SWSR as dirty, and segregation and recycling roles were associated with uneducated people. As a result, the collectors, including buyers of waste segregated materials, were despised in the communities where they lived and worked, as one key informant stated below;


*“It is a dirty job; the perception of communities on wastes is also an issue that whoever works on waste, people think they are of low level, like they are failures and no one wants to be associated with that. No one ever says money is dirty but the means of making money affects our attitudes on how to make the money anyways because we are sensitive to what people think about us and what my neighbors will think if I tell them that I am a waste worker.”*
P26: Key Informant Kiteezi

##### Routine and Script

At the community level, the waste management routine began with cleaning and collecting the rubbish, which was later stored in sacks and/or storage bins. The storage bins were popular in the upscale communities but not in the slums. This was mainly due to storage space challenges for the slum dwellers and theft of the bins. The waste was collected until a day pending collection by the designated collection truck. The residents carried the sacks to the collection truck, which was usually in a designated spot since the access roads were narrow due to the congestion and poor housing structure. The residents who were unable to carry their rubbish to the collection truck or who missed the truck on the day of the collection were left struggling with where to dispose of their waste. As a result, the waste was dumped in the neighboring premises and the drainage channels.

##### Norms

SWSR was not yet common practice and normal behavior in the study areas due to limited sensitization and migration of people in and out of the study areas making it hard to keep up with the behavior as indicated below;


*“The way we handle the different types of solid waste generated from our homes is not very good. Solid waste segregation and recycling is not yet a common practice since we have not been sensitized on how to handle it properly. Even if there are some trainings on solid waste segregation and recycling, the places where we stay, are full of people who are mobile and keep on shifting thus making it hard for the replication of the trainings since you can’t keep training individuals. Nevertheless, we need to be trained and sensitized in the way to handle our waste”*
P6: FGD participant Kasubi-Residential

##### Objects and Infrastructure

Segregation containers were mentioned as the necessary infrastructure for proper waste segregation. The lack of these infrastructure components forced some community members to indiscriminately dispose of the solid waste, sometimes leading to foul smells as indicated below;


*“First of all, the capacity to do that, for you to segregate, you must be having the mechanism, you need to have proper containers, and so if you don’t have containers, you cannot segregate. Secondly, you must be having man power to do that on almost every time and the ability to hire somebody to always do that.”*
P19: Key informant government parastatal

The need and desire for waste segregation equipment were mentioned by participants, though they indicated that these were costly for them since they were poor, as indicated below;


*“Poverty, majority of the people would have liked to have extra bins for waste storage and segregation but cannot afford given the cost implication it has on their budgets.”*
P30: Key Informant Kasubi

##### Touch Points

The main communication channels that could be used in promoting environmental and public health activities such as segregation and recycling in the community included; megaphones, loud speakers, community meetings, and door-to-door visits. Some communities were well organized and had structures and systems that allowed the proper flow of information. The market communities, for example, had leadership structures that could pass important information on to their community members. Door-to-door meetings following COVID-19 Standard Operating Procedures were much more valued during the COVID-19 pandemic since meetings were prohibited.


*“In this market we have a department that is in charge of media and any other communication that people need to receive and in this we make use of our big megaphone installed up there which reaches a bigger population. Therefore, any communication regarding the need, time and the like of doing things as far as sanitation is concerned is always handled and communicated timely by the concerned persons since we have all the departments.”*
P23: Key Informant Market leader

#### 3.2.5. Context-Related Drivers

The presence of companies that dealt in the recycling of different types of waste was seen to influence segregation in the community by providing training on waste segregation and buying the segregated materials. The study revealed that the sought-after waste did not become littered around the community and even when it was littered, there was always someone looking for it. Case in point is metallic wastes and plastic bottles that were collected by children and groups of people to sell to the plastic producing companies for recycling and scrap dealers for cash. It is a case of obtaining cash for the waste. Participants however indicated that the recycling companies were few and not very aggressive in buying off the segregated waste. This finding was revealed by different participants as below;


*“...I know Usafi market, the food section has done so well at diverting the green wastes. They do it perfectly, they employ someone to make sure that no polythene bags go in and the Coca-Cola company is also doing well in recycling, they are trying to buy back all the plastics they put out. But of course, they don’t do it very aggressively, they do it slowly, they are not pushing it too much.”*
P26: Key informant Kiteezi


*“That issue of sorting rubbish, we tried to do it since we got trainings from different organizations. Some people have tried sorting plastic bottles. We had got lucky when coca cola, Pepsi Company and Riham companies intervened to get us places to dispose the plastics. But this other decomposing rubbish was collected, especially during this time of COVID and people started making briquettes out of them. However, those are just a few of them. So, it needs a lot more effort and I request more organizations to come out and train the people. We can reduce the garbage if they train more people.”*
P35: participant FGD Kamwokya

Respondents reported appreciating the value of waste through the support of non-government organizations (NGOs) and community-based organizations (CBOs). Through such organizational support, their children learnt recycling skills, which keep them busy, and recycled products such as briquettes and door mats were used at home while some were sold for money, as mentioned below;


*“People are learning to see the values of garbage. They are seeing waste as money e.g., bottles and AMREF is giving money to women who do recycling. They make table mats, they make tables as I told you. They also use straws to make mats. These are some of the good behavior, which some NGOs are promoting like AMREF which I said is giving them funding and these activities are now keeping children busy. You find most of them busy making briquettes for fire and even they sell”*
P1: Key Informant Kawempe

The study areas had community champions and groups specifically charged with maintaining hygiene in the areas. Some of these champions were local leaders in the community who emphasized discipline and responsibility among their people.


*“To add on what has been said, we have some people that are champions in the general cleanliness. Almost all the villages in Kamwokya, the chairpersons sat and chose the champions of hygiene and these are the ones that should be given any information concerning the hygiene of the community.”*
P35: participant FGD Kamwokya

By the time of the study, waste segregation was not yet a policy in Uganda; hence, people were not mandated to do it, and those that were involved in segregation performed it at will and convenience. Since segregation was not yet a policy, labelled waste sorting bins were not freely provided in the communities by the government and the market for the waste products was not regulated. More so, waste segregation was seen to be a tiring and expensive activity requiring space and different labelled waste bins as illustrated below;


*“People can’t sort the waste because it is not yet a policy. It is a by the way that is why it is done in places of Kololo. First of all, sorting is tiresome and they have nowhere to keep it. The containers are expensive and you are telling the person to keep garbage for three days. Even the sorting material should be labelled properly and given for free alike anywhere in the world. Garbage containers are given by municipalities, companies or government. So, minus that don’t even think of any intervention.”*
P7: Key informant, private solid waste collection company

There were general solid waste ordinances and by-laws in Kampala, which some respondents believed, if enforced, could bring about a positive change in terms of solid waste management. However, challenges were cited in enforcement, manpower and resources as indicated below;


*“Secondly, we need to ensure proper enforcement of the law, there are byelaws and ordinances of solid waste management but those responsible especially KCCA, if they are very serious, they can wipe out all bad vices in the city. The laxity may be due low capacity technically and have no man power or money to be everywhere but even the little they have if they were deploying it well, it could achieve a lot.”*
P19: Key informant, government parastatal

Some key informants mentioned political interference whereby leaders greatly influenced the community’s willingness to make a financial contribution to waste collection and management in their areas. Additionally, it was also reported that during political campaigns, communities were normally promised free waste collection services, which never came to pass. Even though waste management companies desired to fix the prices of garbage collection, it was reported that politicians always told their electorates that this was supposed to be a free service, as indicated below;


*“The main challenge especially in Kampala is political interference, sometimes we put ordinance but politicians come in and interfere with everything in the community... Politicians usually undo this work of community sensitization on proper waste management by giving their other views such as promising free garbage collection by government which never comes to pass.”*
P1: Key informant Kawempe

Lack of commitment to solid waste management by some stakeholders was mentioned as a factor leading to reduced manpower for improved waste management, as stated below;


*“Another challenge we have is that the Ministry Z is silent about solid waste management. Because we have an environmental health division but they are a bit quiet on solid waste management. You talk about excreta management that one is very loud, they bring so many projects e.g., CLTS, SAN marketing, PHAST etc. But that component of solid waste is not so much pronounced then number two; we tend to put emphasis on hardware as far as solid waste management is concerned, we look at picking and taking but that people are not aware, we should first make sure that people know the bad part of the waste, if I accumulate garbage, what are the health issues.”*
P1: Key informant Kawempe

## 4. Discussion

This was a qualitative study exploring drivers to SWSR in Kampala using the BCD theory. SWSR was practiced to a small extent in the community for some waste types facilitated by the anticipated monetary rewards, availability of market for the finished products, and social networks in the community. The key barriers to SWSR were limited space for citing waste segregation containers, limited knowledge of the benefits, limited skills, lack of segregation containers, low value for some waste types and a poor political, administrative and legislative environment.

The physical environment was not conducive to supporting SWSR with limited space where to place containers and the presence of few skips in the community. This is because our study focused on informal settings, which are normally characterized by congestion and poor service delivery. This finding is in agreement with previous studies where limited space, bins and the distance from homesteads to recycling facilities or centers were reported as major factors that prevented community participation in segregating solid waste [13,28,29]. The lack of an enabling physical environment for waste segregation continuously promotes indiscriminate disposal of waste leading to several public health hazards.

Our study revealed the role played by social capital and networks in promoting SWSR in the community. Increased appreciation of the benefits of SWSR may have prompted some people to seek mentorship and training in the same from fellow community members. Currently, there is no institution that is offering training in SWSR; only short-term training is offered in workshops and with a limited number of people. Peers and friends have therefore been the closest source for such skills and may probably be also because of fear of being judged by neighbors if not segregating and recycling. Our findings are in agreement with previous studies where segregation and recycling of waste were influenced by social pressures, neighbors and friends [30,31,32]. The impact of social networks presents a good opportunity for peer-to-peer influence bringing about wide coverage of impactful sensitization.

Most FGD participants expressed limited knowledge on how to segregate and recycle waste and the potential benefits as a key barrier. This may possibly be because SWSR is not yet largely explored by the local population, and there is no institution offering formal training. As a result, few people have the capacity to segregate and recycle waste. Findings in a similar setting in Uganda and elsewhere in Thailand, India and Palestine also indicated a lack of knowledge, sensitization and training on solid waste management as barriers for SWSR [13,33,34,35]. Knowledge is, therefore, a pre-requisite for improved SWSR coupled with other interventions.

Currently, SWSR is practiced for only a few types of waste, which are perceived to have a fairly good monetary value and a ready market. These included food peelings, plastics and polythene bags. The major motive was financial gains. This may be attributed to the process of SWSR, which is considered dirty and tiresome hence one needs compensation for the time invested in the process. A study in a similar setting in Uganda revealed the same where segregation was carried out for only banana peelings and food leftovers for feeding animals [29]. Studies elsewhere have also shown the main motive for SWSR to be for financial gains as opposed to other benefits [13,33]. A study in China, however, showed otherwise, where monetary incentives were not a motivating factor for SWSR [36]. This may have been because the monetary incentives were too low. High monetary incentives have been found to increase the rate of SWSR [37,38,39]. Therefore, in communities with prevailing markets for waste products, hinging campaign messages for SWSR on fair financial incentives is key for the success of the segregation and recycling programs.

The lack of waste segregation containers and infrastructure was also indicated as a challenge and barrier to SWSR. Where such infrastructure was absent, poor SWSR practices were reported, and this was in line with a study in Palestine [33].

Participants indicated the need for different containers that are labelled for different wastes being sorted. Provision of the waste containers for segregation is part of creating an enabling environment for behavior change and sustainability. Different bins that are labelled are believed to remind community members of the need to segregate waste at their disposal. Our finding is similar to previous studies conducted elsewhere, which showed that availability of waste and recycle bins motivates people to segregate and recycle waste [13,33,38,40,41]. On the contrary, a study in Thailand showed no statistical significance between the availability of recycling bins and segregation at source [42].

Our study findings revealed a lack of political will in supporting proper SWSR as a critical barrier. A few policies and by-laws were reported, but unfortunately, implementation was a challenge, especially due to political interference and lack of an implementation strategy. This was expressed in terms of political interference in solid waste management and the lack of a clear legislative framework to support SWSR. Waste segregation is not yet reflected in policy and hence practiced at will, which also limits enforcement. In addition, some relevant stakeholders were cited as not very active in supporting waste management best practices, despite having its practice and service under their docket. This is similar to study findings elsewhere, where the legislative framework is unclear, and there are no political commitments [40,43]. In some countries where there is political will, with governments providing incentives for SWSR and penalties for non-compliance with set standards, recycling has greatly improved [36]. Lack of a political will demotivates the implementers and denies an opportunity to the community to do the right thing for a better environment.

Unique to our study was the use of the BCD theory to systematically explore the drivers of SWSR. This theory considers SWSR as a behavior that is not independent of the person’s environment, brain, and context. The model has previously been used for studying WASH behaviors, nutrition, food hygiene and other behaviors [22,23,24,25]. Unlike other behavior change models, such as the COM-B model [44], the BCD model helps to understand factors related to the behavior setting and context. In this study, the model was found to be useful in understanding the drivers of SWSR behavior and can therefore be adopted by other researchers and public health practitioners in developing interventions for improving solid waste management. However, our study only utilized qualitative methods of data collection; hence, we were unable to quantify the practice in the community. The study findings however give a deeper understanding of the situation at hand.

## 5. Conclusions

Whereas SWSR was not largely practiced in the study slums, it is feasible in these slum settings, especially if interventions are focused on increasing household incomes and promoting safe environments. The major drivers of recycling were mainly related to the physical environment, especially the space where SWSR equipment is located, and social networks, which were associated with enhancing awareness of benefits. Moreover, brain-related factors such as awareness of the benefits and stakeholders, and motives especially financial rewards and the need for a clean environment, were reported to be key in the success of segregation and recycling programs. It was clear, that though SWSR were not habitual, sustainable interventions and adoption of the behaviors require very good political, administrative, policy and legal structures and support, which were unfortunately not fully available by the time of the study.

There is, therefore, a need to sensitize communities on the environmental and public health benefits of SWSR. In sensitizing slum dwellers, financial rewards and environmental safety should be used as the major motives. The interventions in slum settings should go beyond creating awareness to ensure that the necessary infrastructure such as segregation bins, sacks as well as appropriate and accessible collection points are provided. Moreover, sustainable markets for solid waste segregated and recycled products should be created and regulated by the government. Public health officers will need to work with other stakeholders in creating linkages between SWSR communities of practice and buyers of the segregated and recycled waste products. Public–private partnerships will be critical in promoting SWSR though this requires a good policy and regulatory framework, which is currently missing.

## Figures and Tables

**Figure 1 ijerph-19-10947-f001:**
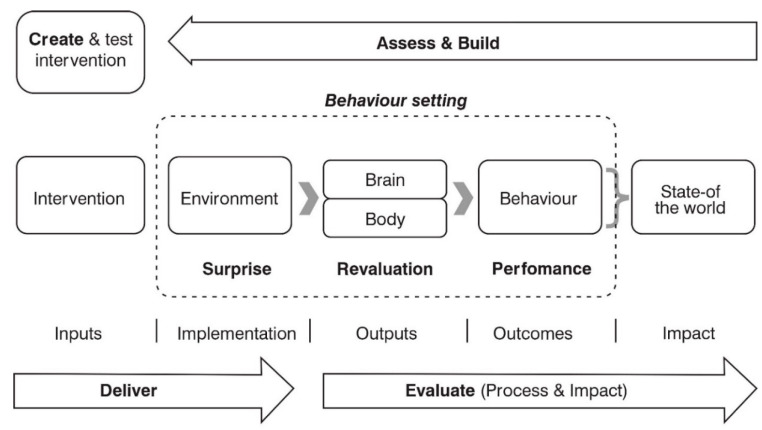
Behavior centered design Model [15].

**Table 1 ijerph-19-10947-t001:** Definitions of solid waste segregation and recycling determinants adapted from the Behavior centered design checklist [26].

Behavioural Determinants Defined by the BCD Framework		Questions that Were Considered
Environment	Physical	What is the physical setting like? What things in the physical environment enable or prevent SWSR? Is SWSR affected by the physical or built environment including climate/weather and geography?
	Social	Does the social environment *(relationships, networks and organisations)* affect SWSR?
Brains	Executive	Do the target communities know the *need and benefits for SWSR* and how it should be performed?
	Motivated	Is SWSR rewarding? Does *disgust* (the desire to avoid cues to sources of infection), *affiliation* (the desire to fit in with others) and *nurture* (the desire to care for your child).
	Reactive	Is SWSR habitual? Did it become a norm?
Body	Characteristics	Are there Socio-demographic characteristics (gender, wealth, age, education and employment etc) that influence SWSR?
	Senses	The sensory perceptions (smells of solid waste and sight) that may influence SWSR?
	Capabilities	Are there individual skills required to segregate and recycle waste? Do individuals perceive themselves to have the ability to segregate and recycle waste?
Behaviour settings	Stage	Where does SWSR take place? Are there concerns related to the specific physical spaces where SWSR take place?
	Roles	What is the role played by the community in SWSR and how does it relate to roles played by the authorities in charge of waste management? Are there ways in which an individual’s role, identity or responsibilities influence their SWSR practices?
	Routine and script	How does the daily routine of activities undertaken by the community influence SWSR?
	Norm	What SWSR behaviour is the community expected to have? Is SWSR a common practice in the community (descriptive norm); is SWSR part of some individual’s role and normal behaviour (personal norm); is SWSR socially approved of (injunctive norm); and is SWSR practiced by individual’s ‘valued others’ (subjective norm)
	Objects and infrastructure	Are there objects needed by the community to do SWSR available? What infrastructure needs to be in place to perform SWSR?
	Touch points (Communication channels)	Are there mechanisms through which community members are receive information/messages related to SWSR? Which mechanism are these?
Context	Programmatic, political, economic, social, and legislative framework	Are there active SWSR programs in the study area? Are there political or historical events that have influenced SWSR programs in this area? Are there laws and policies about SWSR? Are there opportunities and gaps in the legislation?

**Table 2 ijerph-19-10947-t002:** Socio-demographic characteristics of Focus group discussions’ participants.

Variable	Category	Frequency (*n* = 63)	Percentage (%)
Age (years)			
	20–29	11	17.46
	30–39	28	44.44
	40–49	16	25.40
	Over 50	8	12.70
Sex			
	Male	27	42.86
	Female	36	57.14
Highest level of education			
	No formal education	1	1.59
	Primary	26	41.27
	Secondary	32	50.79
	Tertiary	4	6.35
Religion			
	Anglican	25	39.68
	Catholic	15	23.81
	Muslim	9	14.29
	Pentecostal	14	22.22
Occupation			
	Employed	12	19.05
	Self-employed	8	12.70
	Business	29	46.03
	Casual labourer	5	7.94
	Student	2	3.17
	No employment	7	11.11

**Table 3 ijerph-19-10947-t003:** Linkages between codes, subthemes and Themes.

Themes	Subthemes	Codes
Environment related drivers	Physical environment	The siting and construction of houses in the study areas is poorly planned
		The houses are so congested for SWSR to happen
		No physical space for putting waste segregation equipment
	Social environment	Encouragement to segregate and recycle as a result of peer influence
		Waste segregators and recyclers influence some people staying in their geographical areas.
		Collective actions which promote learning and behavior change
Brains related drivers	Executive brain	Awareness on issues related to solid waste segregation and recycling
	Motivated brain	Appropriate waste segregation and recycling is associated with rewards such as money, a clean and safe environment
Body related drivers	Socio-demographic characteristics	Segregation of waste can be easily performed in upper income residences, which have space and better roads
		SWSR can easily be performed by institutions and middle-income people.
	Human senses	Smells hinder people from segregating and recycling waste.
	Capabilities	Waste segregation and recycling requires special skills
Behaviour setting drivers	Stage	Presence of bins for waste segregation
	Roles	Individuals (especially women and children) play a critical role in waste segregation and recycling
		Companies (waste companies and NGOs) play a role in waste segregation and recycling
	Routine and script	The SWSR routine involves cleaning and sorting waste into useful streams
	Norms	SWSR is not norm
	Objects and infrastructure	SWSR requires infrastructure such as waste bins
	Touch points	SWSR requires behavior change communication through mega phones, loud speakers, community meetings, and door to door visits
Context related drivers	Policy framework	No policy and by-laws on SWSR
	Stakeholders	Stakeholders critical in SWSR

## Data Availability

Data are available upon reasonable request from the corresponding author.

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
