# Peer review of "Drivers of Solid Waste Segregation and Recycling in Kampala Slums, Uganda: A Qualitative Exploration Using the Behavior Centered Design Model"

_ijerph, 2022, doi:10.3390/ijerph191710947_

Round 1
Reviewer 1 Report
In my opinion, this manuscript is interesting topic, however lacking in terms of scientific contribution. I suggest authors to improve results & discussion section, also most important to add some research synthesis and come up with some specific recommendation.
Author Response
Thank you very much Sir/Madam for the comments. We have carefully looked at the comments and attached here is the point by point response to the comments.

Reviewer 2 Report
In the study, public-private partnerships was promoted as critical in SWSR management though this requires a good policy and regulatory framework, which is currently missing. Solid waste reduction, reuse, and recycling are feasible in slum settings such as those in Kampala, though there is a need to sensitize communities on the environmental and public health benefits of the approach. In sellers on aspects of solid waste reuse and recycling, financial rewards and environmental safety should be used as the major motives. However, SWSR interventions in slum settings should go beyond creating awareness to ensure that the infrastructure is provided, and sustainable markets for solid waste products are created. Besides, public health officers will need to work with other stakeholders in creating linkages between communities of practice (SWSR) and buyers of segregated and recycled waste products.
This is an interesting paper and should be published after minor revisions as below.
General Remarks
- Abstract should be supplemented with the most important research results
- Conclusions are too long and should be decreased by 25%. These should include a discussion of the research results.
- Please do not use abbreviations in the Abstract or provide the full name.
Detailed remarks
1. Line 95 - …”and key informants’ interviews (KIIs)”…. check, please
2. Figure 1- In the title should be the full name of BCD
3. Table 1 - In the title should be the full name of BCD
4. Table 3 – the title is too general, change it, please
5. Table 2 should be self-explaining. Please add proper information.
Author Response

(The authors gave the same response as above.)

Round 2
Reviewer 1 Report
In my opinion, the revised manuscript is improved.